

# Continuous measurements of valley floor width in mountainous landscapes

Fiona J. Clubb[1], Eliot F. Weir[1,2], and Simon M. Mudd[2]

[1]Department of Geography, Durham University, UK
[2]School of GeoSciences, University of Edinburgh, UK

**Correspondence:** Fiona J. Clubb (fiona.j.clubb@durham.ac.uk)

**Abstract.** Mountainous landscapes often feature alluviated valleys that control both ecosystem diversity and the distribution of human populations. Alluviated, flat valley floors also play a key role in determining flood hazard in these landscapes. Various mechanisms have been proposed to control the spatial distribution and width of valley floors, including climatic, tectonic and lithologic drivers. Attributing one of these drivers to observed valley floor widths has been hindered by a lack of reproducible, automated valley extraction methods that allow continuous measurements of valley floor width at regional scales. Here we present a new method for measuring valley floor width in mountain landscapes from digital elevation models (DEMs). This method first identifies valley floors based on thresholds of slope and elevation compared to the modern channel, and uses these valley floors to extract valley centrelines. It then measures valley floor width orthogonal to the centreline at each pixel along the channel. The result is a continuous measurement of valley floor width at every pixel along the valley, allowing us to constrain how valley floor width changes downstream. We demonstrate the ability of our method to accurately extract valley floor widths by comparing with independent Quaternary fluvial deposit maps from sites in the UK and the USA. We find that our method extracts similar downstream patterns of valley floor width to the independent datasets in each site. The method works best in confined valley settings and will not work in unconfined valleys where the valley walls are not easily distinguished from the valley floor. We then test current models of lateral erosion by exploring the relationship between valley floor width and drainage area in the Appalachian Plateau, USA, selected because of its tectonic quiescence and relatively homogeneous lithology. We find that an exponent relating width and drainage area ($c_v = 0.3 \pm 0.06$) is remarkably similar across the region and across spatial scales, suggesting that valley floor width evolution is driven by a combination of both valley wall undercutting and wall erosion in the Appalachian Plateau. Finally, we suggest that, similar to common metrics used to explore vertical incision, our method provides the potential to act as a network-scale metric of lateral fluvial response to external forcing.

## 1 Introduction

Many readers of this journal will have heard, in a classroom environment, the received wisdom that glacial valleys are shaped like the letter "U" and fluvial valleys are shaped like the letter "V". You may have said this out loud to a room full of students. This description is not bad if we are forced into a letter-based identification system for geomorphic features. However, if you move a few kilometres downstream of the channel head in a mountain river you are unlikely to find the river sitting at a



point formed by the bottom of two straight hillslopes. You will find instead a flat, or almost flat, surface that we might call a floodplain, a combination of floodplains and terraces, or simply a valley floor. Valley floors, in addition to being the flattest parts of mountain environments, are often the most fertile (e.g., Tockner and Stanford, 2002). Therefore, they tend to be where settlements, vegetation, and farming are concentrated (e.g., Cooper et al., 2003; Thorp et al., 2006; Felipe-Lucia et al., 2014; Tomscha et al., 2017). In the rest of this paper we will refer to the base of fluvial valleys as "valley floors".

Wide alluviated valleys tend to be productive riverine ecosystems (Tockner and Stanford, 2002), forming important seasonal habitats for species such as Pacific salmon (May et al., 2013; Beeson et al., 2018). The width of rivers and their associated valleys also has important implications for flood hazard. Channel width sets the cross sectional area of a river, and therefore the maximum flow discharge that the channel can contain before flooding occurs (Lane et al., 2007; Slater et al., 2015). The width of the valley floor, which we define as the channel plus that of the floodplain and any terrace remnants, controls how 35   confined flood waters are once this channel capacity is exceeded. Despite the importance of valley floors for communities and ecosystems in upland environments, our understanding of what controls the location and width of these valleys is surprisingly poor (May et al., 2013).

   Over longer timescales, river valley floors can widen or narrow in response to changes in climate, lithology, and tectonics (e.g. Brocard and van der Beek, 2006; Schanz and Montgomery, 2016; Langston and Temme, 2019), providing the potential to 40   use valley floor width as a metric to understand how landscapes respond to external forcing. Lateral migration of channels may be an important control on both channel profiles (Finnegan and Dietrich, 2011) and the evolution of drainage networks over geological timescales (Kwang et al., 2021). Some studies have suggested that lateral erosion rates can outpace those of vertical incision (Suzuki, 1982; Cook et al., 2014; Marcotte et al., 2021), yet research into rates and mechanisms of lateral erosion is distinctly lacking compared to vertical bedrock incision.

## 1.1   Controls on valley floor widening

Conceptually, we might expect the width of a river valley floor in an upland landscape to be controlled by the ratio of vertical to lateral erosion. Gilbert (1877) suggested that lateral erosion will become more important than vertical erosion when sediment supply nears transport capacity. Through time, changes in slope, discharge, or sediment supply may result in increasing vertical erosion rates: in this case, the active channel will incise, leaving behind an abandoned bedrock surface with a veneer of alluvial 50   sediment referred to as a strath terrace (Mackin, 1937; Finnegan and Dietrich, 2011). Hancock and Anderson (2002) suggest that, intuitively, valley floor widening can only occur when the channel is in contact with the valley wall. If we imagine a valley floor that is wider than the active channel, then we can hypothesise that the rate of valley floor widening will depend on the ratio of the active channel width to the valley floor width. If the active channel width is close to that of the valley floor width, there will be a greater probability of the channel impinging upon the valley wall, whereas if the channel width is small 55   compared to the valley floor width, lateral erosion will occur less frequently. This ratio can change in one of two ways: either by widening/narrowing the valley, or widening/narrowing the active channel.

   Empirical studies of valley floor width in bedrock systems have suggested a simple power-law relationship between valley floor width and drainage area, as many workers have reported a relationship between valley floor width, lithology, and discharge





(e.g. Snyder et al., 2003; Tomkin et al., 2003; Brocard and van der Beek, 2006; Lifton et al., 2009; May et al., 2013; Schanz
and Montgomery, 2016; Beeson et al., 2018; Langston and Temme, 2019). This relationship takes the form:

$$W_v = K_v A^{c_v}, \tag{1}$$

where $W_v$ is valley floor width, $K_v$ is a coefficient describing the influence of lithology on valley widening, $A$ is drainage
area, and $c_v$ is an exponent describing how quickly valley floor width changes with drainage area. Brocard and van der Beek
(2006) measured valley floor widths and drainage area in the western Alps, and found that $c_v$ generally fell within the range
of 0.3 - 0.4, and $K_v$ varied between 8 - 160 m km$^{-0.8}$ depending on lithology if $c_v$ was set to 0.4. Tomkin et al. (2003)
measured valley floor widths for the Clearwater River, Washington, and found higher $c_v$ values of 0.76. This model suggests
that increasing downstream discharge is the main control on valley widening, whereas lithology (through $K_v$) influences the
width at any given discharge. Langston and Temme (2019) found that in the western French Alps $c_v$ varied with lithology from
0.099 in the hardest lithologies to 0.42 in the softest lithologies. This suggests that in regions where the valley floor and valley
walls are made of the same material, we may expect channel and valley floor width to increase at the same rate, whereas in
regions where the valley walls are made up of a different material these rates may drastically differ. For example, in a valley
with significant alluvial fill but bedrock walls, the channel may be able to widen rapidly with changes in discharge, whereas
the valley floor widening rate will be set by the erodibility of the bedrock walls (Langston and Temme, 2019). Langston and
Tucker (2018) developed a numerical model simulating lateral erosion in bedrock channels, which allows for two different
mechanisms of valley floor widening: one in which the entire valley wall must be eroded for lateral migration to occur, and
one where undercutting and slumping can result in more rapid lateral erosion. Using the undercutting-slump mechanism,
which may be suitable for regions with soft, easily erodible bedrock (Johnson and Finnegan, 2015), they found a power-law
relationship between valley floor width and drainage area with $c_v = 0.46$ and $K_v = 0.16$.

A simple model of valley floor width based on drainage area and lithology alone lacks consideration of several factors which
may influence valley floor width (Figure 1). Along with discharge, sediment supply is an important parameter that can either
cause or hinder valley widening. If more sediment is supplied to the channel than can be transported (for example, through
collapse of the valley side wall), this sediment can act to protect the valley walls and decrease the rate of lateral incision
(Malatesta et al., 2017). Valley wall height is therefore an important parameter in setting lateral migration rates of the channel:
sediment needs to be eroded from the entire height of the valley or the channel for the valley floor width to increase (Bufe
et al., 2019; Tofelde et al., 2019). Recent analytical models have suggested that in systems dominated by abrasion, the rate
of particle impacts and the volume detached with each impact is important in setting lateral erosion rates (Turowski, 2020; Li
et al., 2020). This suggests that variations in lithology between the valley walls and sediment within the channel should affect
valley floor width. Baynes et al. (2020) examined channel width and sediment lithology in the Rangitikei River, New Zealand,
and found that channels containing sediment derived from resistant greywacke were up to an order of magnitude wider than
those without. While this study looked at channel width rather than valley floor width, the same mechanism may be expected
to control valley floor width over longer timescales than active channel adjustment. Cook et al. (2014) observed that valley



widening in an active reach in Taiwan was concentrated along reaches where the channel was curving, and limited where it was straight, suggesting that abrasive particles, directed at valley walls, were the main driver of the widening process.

The role of tectonics is not included within simple scaling relationships between valley floor width and drainage area, and yet
rates and spatial patterns of uplift have been shown to correlate with valley floor width changes in tectonically active regions. Non-uniform patterns of uplift affect channel slopes: faster flow in steeper channel reaches results in the channel occupying a smaller cross section, suggesting that channel slope and therefore uplift should be a key control on valley floor width (e.g. Finnegan et al., 2005). Whittaker et al. (2007) investigated the Rio Torto in the central Italian Apennines which crosses an active normal fault. They found that the ratio between the channel width to the valley floor width increased directly upstream of the
fault strike as the wide, partly alluviated upstream valley transitioned to an incised gorge. To take into account the influence of channel slope on valley floor width, Brocard and van der Beek (2006) suggested an alternative model of valley floor width evolution that assumes that valley floor width is set by the frequency of strath terrace erosion in transport-limited reaches. They suggest that valley widening occurs when erosion is high enough to rework all the alluvial fill in the valley down to the bedrock strath underneath. The frequency of erosion of the strath is set by the ratio of current erosion in the reach, which is
transport-limited, to a hypothetical maximum erosion which is detachment-limited. This model is consistent with the findings of Cook et al. (2014), who showed erosion of valley walls in an active gorge in Taiwan slowed to almost zero subsequent to deposition of alluvium on the valley floor.

Alongside tectonics, the glacial history of a region is an important parameter that may control valley floor width. In post-glacial landscapes, valley floor width may be preconditioned by prior glacial erosion leading to valleys which are much wider
than the active channel width. Glacial landscapes may also have complex patterns of meltwater discharge and sediment supply (e.g. Dadson and Church, 2005; Brardinoni et al., 2018), as well as base-level changes which have been shown to influence upstream patterns of valley floor width (e.g. Gran et al., 2013). Current models of fluvial erosion and lateral migration are not generally developed with high-latitude, post-glacial systems in mind, despite their prevalence over large regions of the Earth's surface.

Proposed models of valley widening have made various testable hypotheses about how the width might vary as a function of environmental factors. For example the models of Langston and Tucker (2018) suggest that different valley widening processes result in different values of the exponent $c_v$, and the model of Brocard and van der Beek (2006) suggests that topographic gradient should influence the width of valley floors in a predictable manner. Cook et al. (2014) suggested that the width of valley floors should increase where rivers have higher curvature, on the basis of oblique particle collisions with valley walls.
Lancaster (2008) hypothesised that width will increase in areas with greater probabilities of debris flow sources in headwater areas. In any of these cases, testing of hypotheses relies on accurate measurements of the width of the valley floor. Here we concentrate our efforts on a method that allows us to reproducibly measure the width of the valley floor, and use this method to explore the value of $c_v$, which has been linked to lateral erosion process, in a landscape where tectonics and lithological heterogeneity are unlikely to play a role.





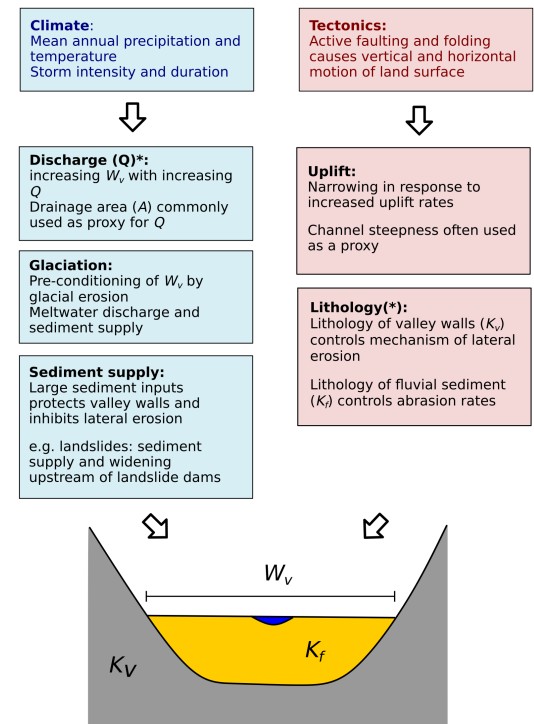

**Figure 1.** Conceptual model highlighting key controls on valley floor width ($W_v$). Climate affects river discharge, sediment supply, and the extent of glaciation within catchments. Tectonics controls uplift rates and lithology by juxtaposing contrasting lithological units. The influence of each factor on valley floor width is highlighted in the respective box. Factors labelled with an asterisk(*) are those included within existing models predicting valley floor width.

## 1.2 Methods for calculating valley floor width

Most studies which have constrained valley floor width in mountain landscapes have used either field measurements (e.g. Lancaster, 2008; Lifton et al., 2009), hand-mapped valley floor widths from topographic maps (e.g. Brocard and van der Beek, 2006) or digital elevation models (e.g. Gran et al., 2013; Schanz and Montgomery, 2016; Langston and Temme, 2019). These methods provide good constraint on valley floor width over small scales, but the time involved in either collecting field measurements or hand-mapping widths limits our ability to collect these measurements over large spatial scales. Measurements are generally taken hundreds of metres apart, meaning we lack a continuous dataset of valley floor width to compare to channel slope measurements, for example. Furthermore, different workers may have their own definition of what constitutes the edge of the valley, or it may be difficult to objectively identify. In the field, the presence of dense vegetation can make valley floor width hard to measure. Hand measurements from maps or shaded relief rasters can result in ambiguous results where the edge of the valley is not sharply defined, or where there are significant anthropogenic modifications of the valley floor.



There are relatively few attempts to automate the extraction of valley floor widths from digital topography. Daxberger et al. (2014) presented a toolbox which allows the calculation of valley cross sections to identify topographic symmetry: their tool identifies the valley floor using a "flatness" threshold, where the valley floor is identified as regions of the cross section with a slope change of < 12.5%. Zhao et al. (2019) developed an automated technique for calculating valley floor width by taking the
DEM-derived trunk channel and finding break-in-slopes perpendicular to the flow direction. This algorithm is similar to the approach we employ here, and is available as an ArcGIS plugin. However, their method relies on a sharp transition between the valley floor and surrounding hillslopes, as it is only based on slope rather than elevation compared to the channel. Zhao et al. (2019) suggest their method is more effective in U-shaped valleys rather than V-shaped which are more common in fluvially-carved landscapes. Khan and Fryirs (2020) developed a semi-automated GIS tool that delineates valley bottom polygons and
calculates valley floor width from valley cross sections. This tool needs manual processing and GIS analysis to extract valley floor widths. Zhu et al. (2021) developed a valley finding component as part of a swath mapping tool, but this method is based on a topographic position index that includes the lower part of the hillslope and does not isolate the valley floor. Hilley et al. (2020) presented a curvature-based approach, in which valleys are identified using the scale at which the principal curvature is minimised across a valley cross section. They found this method can distinguish between V and U-shaped valleys, but noted
widths extracted using their method are generally under-predicted compared to manual measurements.

In this contribution, we present a new method for continuously measuring valley floor width from DEMs. Our method is unique in that it is available as part of an open-source topographic analysis software package (LSDTopoTools), and it allows the extraction of the valley centreline to account for meandering systems. Our method allows measurements at every pixel downstream, giving a dataset of how valley floor width varies downstream along valley profiles and across landscapes. We
demonstrate the ability of our method to extract reliable valley floor widths by comparing with Quaternary superficial deposit maps from the UK and the USA, which map out the valley floors in detail. We then explore downstream changes in valley floor width across the Cumberland and Allegheny Plateaus to the west of the Appalachian Mountains to test the hypothesis that valley floor width should scale as a power law function of drainage area.

## 2 Automated extraction of valley floor widths

We build on the technique for automatically identifying floodplains and terraces presented in Clubb et al. (2017). This method identifies floodplain and terrace pixels by calculating two metrics for every pixel in the DEM: the elevation of the pixel compared to the nearest channel, and the local slope. The channel network for the DEM is defined by extracting channel heads using the techniques outlined in Clubb et al. (2014), and flow routing using a steepest descent algorithm from each channel head. For each pixel, these elevation and slope metrics must both be below a defined threshold to be identified as part of
the floodplain or terrace (i.e. a pixel must be relatively flat and near to the elevation of the modern channel). Thresholds for elevation above the channel and local slope can either be calculated statistically using quantile-quantile plots of the distribution of slope and elevation across the landscape, or they can be set manually by the user. The second option is recommended in





lower-relief landscapes where there is less contrast in slope and elevation between surrounding hillslopes and the valley floor. This leads to an integer classification of the raster into valley floor pixels (1), channel pixels (2), or non-fluvial pixels (0).

In some landscapes, particularly those with significant anthropogenic modification of the valleys floors or noise in the topographic data, the initial extraction of the floodplain leads to isolated holes of non-valley floor pixels along the valley. This can confound the extraction of automated valley floor width. We therefore include an option to fill in these holes in the valley floor using a flood filling algorithm implemented within OpenCV (Bradski, 2000).

     Following extraction of floodplain or terrace pixels, we extract the longitudinal profile along which valley floor width will be

measured. This is done using one of two methods. For regions where the river does not significantly meander within its valley, or for coarser resolution DEMs where flow paths tend to be less variable, we extract the steepest descent flow path from a user-defined upstream point on the channel to the outlet of the DEM. However, in some cases (such as where the river meanders within its valley), the steepest descent path may not align with the overall valley trend. In this case, valley floor widths can be over-estimated if extracted from the steepest descent trace. We therefore provide an option of extracting the valley centreline

from which to determine valley floor width (Figure 2).

     Our method of determining the valley centreline is based on creating an artificial V-shaped valley using the topography of the identified floodplain. We begin by ingesting the valley mask and calculating the distance to the nearest valley edge pixel for every pixel identified as belonging to the valley. For brevity we will call these "bank" pixels. The algorithm does this by searching a widening radius until it finds the nearest pixel not in the mask. For every pixel in the valley mask, the elevation of

the nearest bank pixel is recorded along with the distance to the bank. We then run a moving window over these valley pixels to find the minimum bank elevation within the window. The moving window radius is set to be on the order of the width of the valley, so the resulting pixel values are entrenched relative to the original bank elevations. We then subtract elevation from this minimum bank elevation mask: the elevation subtracted is calculated by multiplying the distance from the bank with a scaling factor. This means the largest subtracted values are in the middle of the valley, and the valley forms a trough that is roughly

triangular in cross section. We then overwrite the valley in the original DEM with this "trough" mask. This is then carved and filled repeatedly, using the algorithm of Lindsay (2016), with the trough subtracted on each iteration until there is a single carved centreline. We favour this approach to, for example extracting a floodplain skeleton using computer vision techniques (e.g., Saha et al., 2016), because our method preserves flow routing information of the valley centreline pixels.

     We then move down the centreline using D8 flow routing. We define a pixel spacing, $n$, so that for any given pixel we select

a point $n$ pixels upslope along the centreline and $n$ pixels downslope. Two bearings are calculated: the bearing of the vector starting at the upslope pixel and ending at the valley pixel in question ($b_u$), and the bearing of the vector starting at this valley pixel and the downslope ending pixel ($b_d$). The channel pixel bearing ($b$) is then calculated with:

$$b = 0.5 * (b_u - b_d) + b_d \qquad (2)$$

     All bearings are calculated clockwise relative to North. We then calculate the vector orthogonal to this bearing. We follow

the vector orthogonal to the valley centreline towards both the left valley wall and the right valley wall, and calculate the width



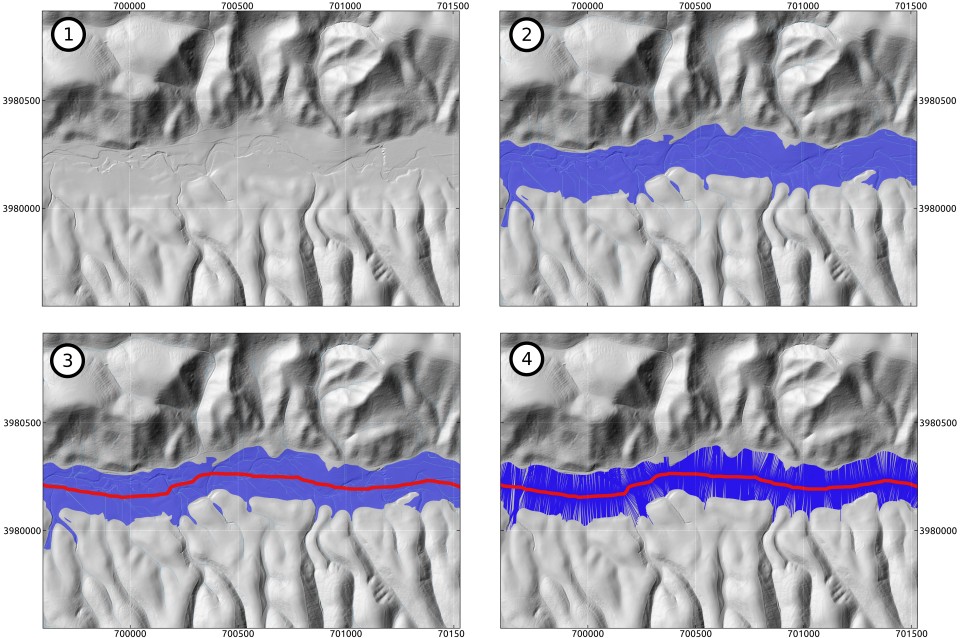

**Figure 2.** Step by step automated extraction of valley floor widths from topographic data: example from 1m lidar DTM from Gabilan Mesa, California. Coordinates are WGS84 UTM Zone 10N. (1) Hillshade showing initial topographic data. (2) Identification of valleys from the DEM (Clubb et al., 2017). Dark blue pixels are identified as floodplain and light blue are identified as the channel network. (3) Extraction of valley centreline (red line). (4) Lines orthogonal to local flow direction are drawn at each point along the centreline until they reach the edge of the floodplain on either side of the valley. Width is measured for each of these lines and can be continuously plotted along the centreline.

of the valley when each of the left and right valley wall vectors enter a pixel that is not in the valley floor mask. An illustration of these vectors are shown in Figure 2.

## 3 Comparison with independent valley floor width measurements

One of the key challenges of testing the efficacy of our method is determining what the 'true' valley floor width should be. Estimates of valley floor width made in the field may be very different between workers, and delineation from datasets such as aerial photography may be hindered by the presence of vegetation or clouds. We therefore take a sedimentological approach to determining independent valley floor width measurements, and compare our automated extraction of valley floor widths to widths mapped from detailed Quaternary superficial geology maps. These maps represent sediment that has been deposited by fluvial processes over the Quaternary period and is therefore likely to represent the active river valley over



geomorphologically meaningful timescales. We tested our method at three field sites located in the UK and the USA: the River Tweed, southern Scotland; Weardale, northern England; and the Russian River, northern California. For the UK sites, we used detailed Quaternary deposit mapping from the British Geological Survey (BGS) at a scale of 1:50,000. For the US site, we used a compilation of Quaternary deposit mapping from the San Francisco Bay region from Knudsen et al. (2000). This dataset compiles Quaternary maps at 1:24,000 and 1:100,000 resolution: we limited our analysis to the lower reaches of the Russian

River where the data are available at the higher 1:24,000 resolution. The Quaternary dataset showed some offset compared to the valleys evident from the lidar DEM: we therefore georeferenced the Quaternary map using control points located at the boundary between fluvial terraces and valley walls.

The BGS Quaternary maps differentiate between sedimentary deposits of glacial, glacio-fluvial, and fluvial origin: we isolated our analysis to fluvial deposits within the valley. That is, we base our valleys floors on the portion of the valley that has

experienced fluvial deposition over the Quaternary period (Figure 3). Similarly, the United States Geological Survey (USGS) Quaternary maps distinguish between Holocene alluvium, river terrace deposits, and non-fluvial deposits such as those from alluvial fans and artificial valley fills. We limited our analysis to deposits of fluvial origin including both terrace and modern floodplain sediment. To compare like-for-like between the automated method and the Quaternary maps, we transformed the vectorised Quaternary deposit data into a binary raster where a value of 1 represented fluvial deposits (both modern alluvium

and river terrace deposits) and 0 represented non-fluvial deposits such as glacial sediments, glacio-fluvial deposits, colluvium, or bedrock. We then applied our technique for calculating valley floor widths described in the Methods section to this binary raster.

## 3.1 River Tweed, Scotland, UK

The River Tweed is a gravel-bedded river with a well-developed valley floor located in southern Scotland, UK (Figure 3),

originating in the Scottish Southern Uplands and draining to the North Sea. The underlying bedrock geology consists of Silurian and Ordovician greywackes, slates, and shales, with extensive glacial deposits (Owens and Walling, 2002). We extracted valley floor widths along the main course of the River Tweed automatically from the Ordnance Survey (OS) Terrain 5 DTM, which has a resolution of 5 m. We found that the downstream pattern of valley floor widths from our automated method compared well to that of the Quaternary maps (Figure 4). The mean valley floor width calculated by the automated method was $344 \pm$

$281$ m, and the mean width calculated from the Quaternary deposits was $312 \pm 235$ m. The mean difference in width between the two datasets was 17 m.

Figure 4 shows that the most significant difference between two datasets occurred in the uppermost 15 km of the study section, with the automated technique detecting a width $\approx 150$ m wider than the Quaternary maps. This difference occurs where the automatically-extracted valley occurs within glacio-fluvial deposits which bound the alluvium on either side of the

channel. Inspection of the hillshade (Figure 5) shows that the widths extracted by the automated method throughout this reach are more consistent with the break-in-slope at the transition between the surrounding hillslopes and the valley floor, suggesting that they would be closer to a typical 'field' definition of valley floor width compared to the Quaternary maps.





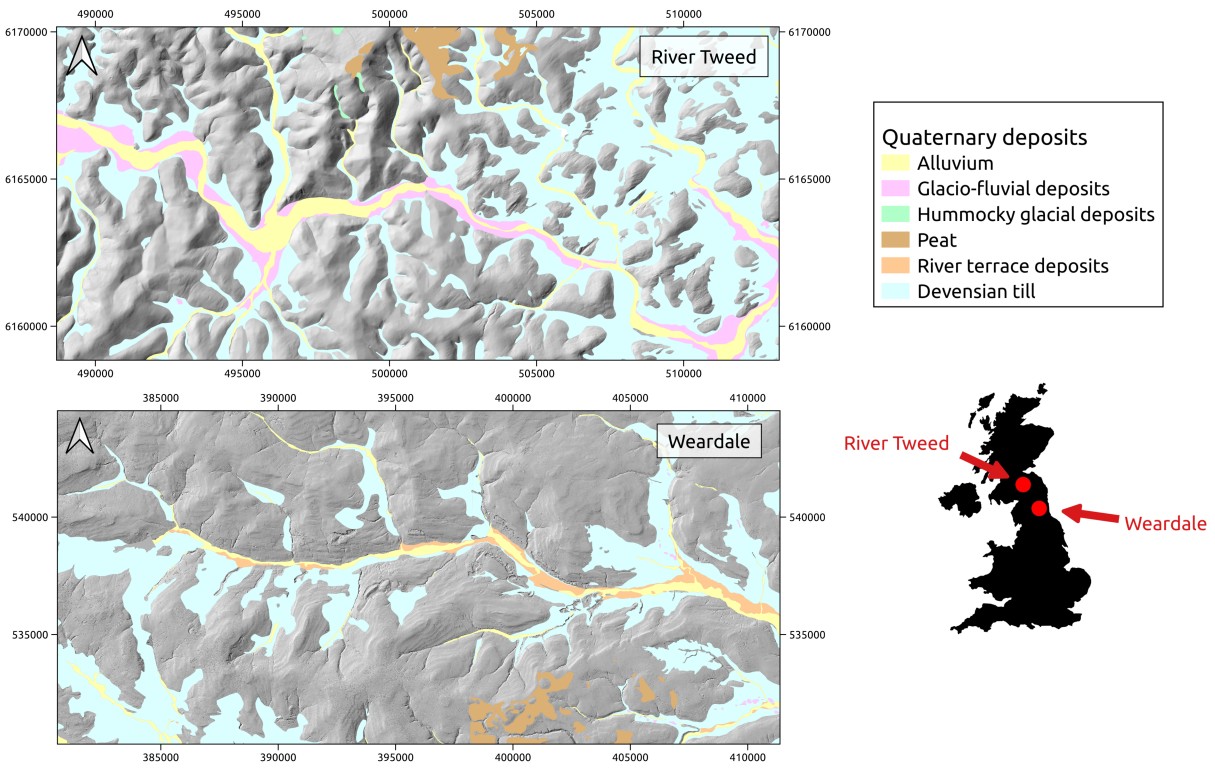

**Figure 3.** Hillshade of the River Tweed and Weardale showing the extent of superficial Quaternary deposits mapped by the BGS. We define the valley as the extent of the alluvium and river terrace deposits combined. The red stars show the approximate location of the sites in the UK. The coordinate system is WGS84, UTM Zone 30N.

## 3.2 Weardale, England, UK

We then tested our algorithm on a more complicated system with preserved fluvial terrace deposits as well as modern alluvium.
We analysed valley floor width along the Upper River Wear in Weardale, a valley with extensive glacial deposits, alluvium and river terraces in the North East of England. Weardale is notable for not being as nice as neighbouring Teesdale. The River Wear is sourced in the English North Pennines, and flows east to the North Sea. Alongside distinguishing between glacial or fluvial origin, the Quaternary deposit maps also separate alluvium and fluvial terrace deposits (Figure 3). We extracted valley floor widths automatically from the 2020 lidar DTM from Weardale compiled by the Environment Agency, which we resampled to
2 m resolution.

Figure 6 shows the pattern of valley floor width downstream from both our automated technique and the BGS Quaternary deposits. We found that the pattern of valley floor widths was similar between the two datasets: the mean width from the





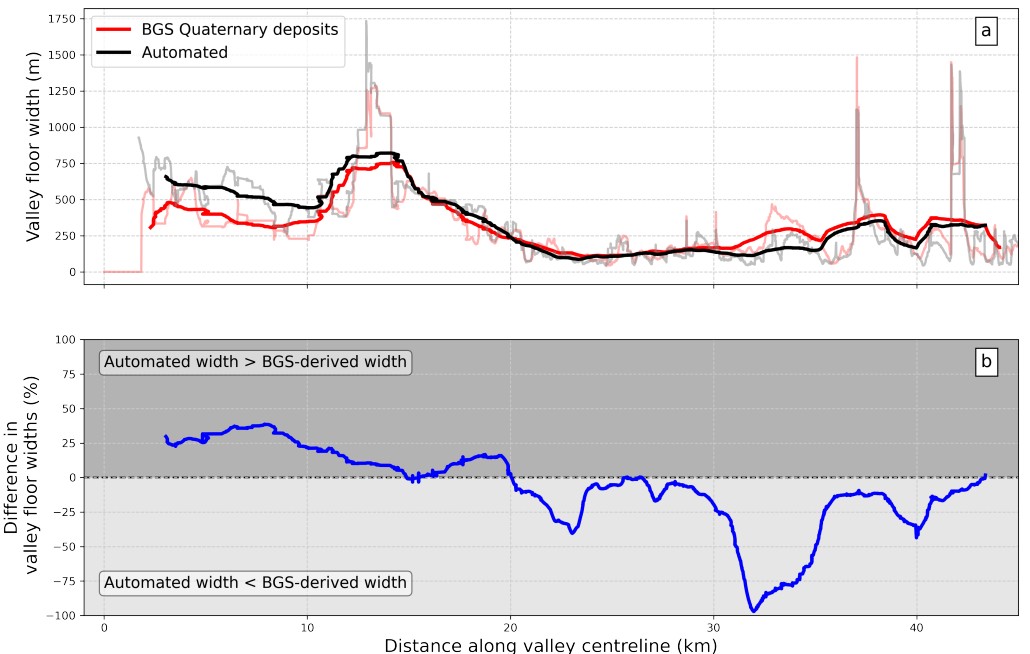

**Figure 4.** Results of valley floor width comparison for the River Tweed. (a) valley floor width from upstream to downstream along the centreline for the automated method (black) and derived from the BGS Quaternary maps (red). The light grey and red lines show the raw data, and the darker lines show a rolling average of width over a 500 m window. (b) Percentage difference between the two datasets along the centreline, where positive values indicate a wider automated width than BGS-derived, and negative values indicated a narrower automated width than BGS-derived.

automated method was 335 ± 224 m, compared to 390 ± 195 m. The mean difference between the two datasets was 27 m. However, the automated widths were ≈ 25% wider than the BGS widths at the upper portion of the catchment, while they were

up to twice as narrow as the BGS widths at 25 - 40 km downstream along the valley centreline. Inspection of the hillshade and Quaternary map reveals that terraces are the result of this difference. The upper part of the catchment contains upper terraces within the valley which are not identified as river terraces on the BGS Quaternary map, but are flat surfaces close in elevation to the modern channel and are therefore picked up by the automated technique. Further downstream, the river terraces are high in elevation compared to the channel, and are therefore missed by the automated technique despite being river terraces on

the Quaternary map. This suggests users should carefully select the thresholds for elevation above the channel in areas with terraces of varying heights. We also include the option of ingesting Quaternary maps for width analysis in our software, so that users can use these as floodplain datasets for width analysis if preferred.





**Figure 5.** Hillshade of the River Tweed showing the valley floor widths extracted using the automated method (black) and those derived from the BGS Quaternary maps (red). The valley centreline is shown in blue. The left zoomed-in region shows an example of where the automated method is wider than the BGS Quaternary, but picks up the break-in-slope transition to the hillslopes. The right zoomed-in region shows an example of where the two datasets are consistent. The coordinate system is WGS UTM Zone 30N.


Earth **Surface**
**Dynamics**
Discussions

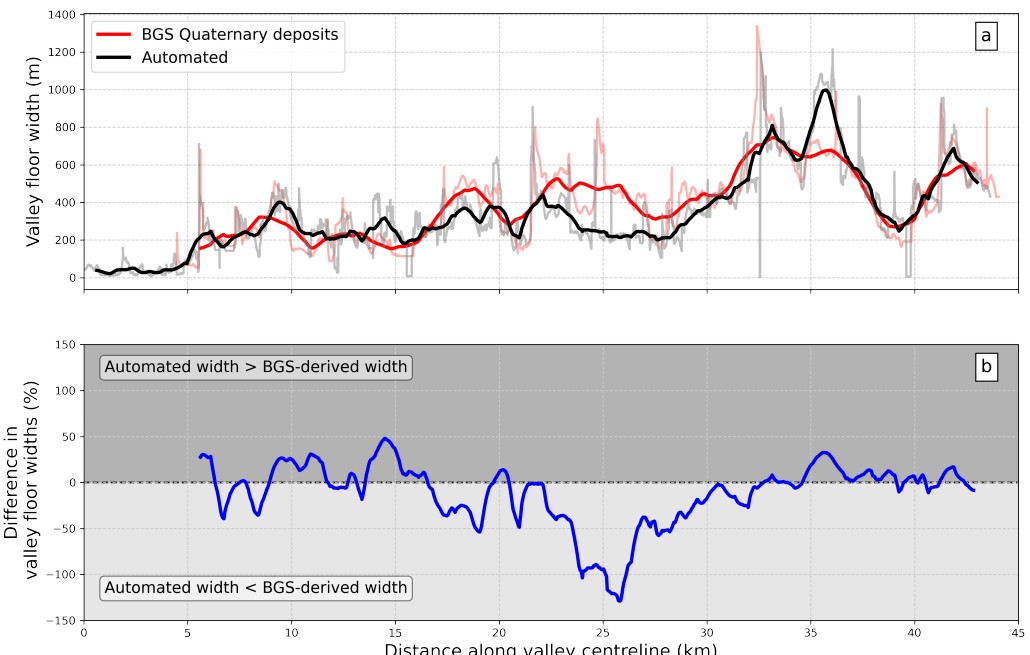

**Figure 6.** Results of valley floor width comparison for Weardale. (a) valley floor width from upstream to downstream along the centreline for the automated method (black) and derived from the BGS Quaternary maps (red). The light grey and red lines show the raw data, and the darker lines show a rolling average of width over a 500 m window. (b) Percentage difference between the two datasets along the centreline, where positive values indicate a wider automated width than BGS-derived, and negative values indicated a narrower automated width than BGS-derived.

We then tested the sensitivity of our method to the resolution of the DEM for the Weardale site by resampling the Environment Agency lidar to 5 m and 10 m, and downloading the 30 m SRTM data for the valley from OpenTopography. This

represents the likely range of DEM resolutions available for the majority of the Earth's surface, as lidar data are increasingly available on regional and national scales and the SRTM 30 m data are freely available on a global scale between 60°N and 54°S. We found that the overall pattern of valley floor width remains consistent up to 30 m resolution, but there are some discrepancies in the valley floor widths calculated at each resolution (Figure 7). For example, at ≈ 36 km along the valley centreline, the 2 m dataset shows widths of up to 1 km, whereas the 10 m, 30 m, and BGS Quaternary deposits indicate valley

floor widths of between 500 - 700 m. Furthermore, the 2 m lidar is able to calculate valley floor widths at the furthest upstream part of the valley, whereas datasets with coarsening DEM resolution can only identify the valleys further downstream. For example, the SRTM dataset only starts identifying valley floor widths at ≈ 4 km downstream along the valley centreline.



Earth **Surface**
**Dynamics**
Discussions



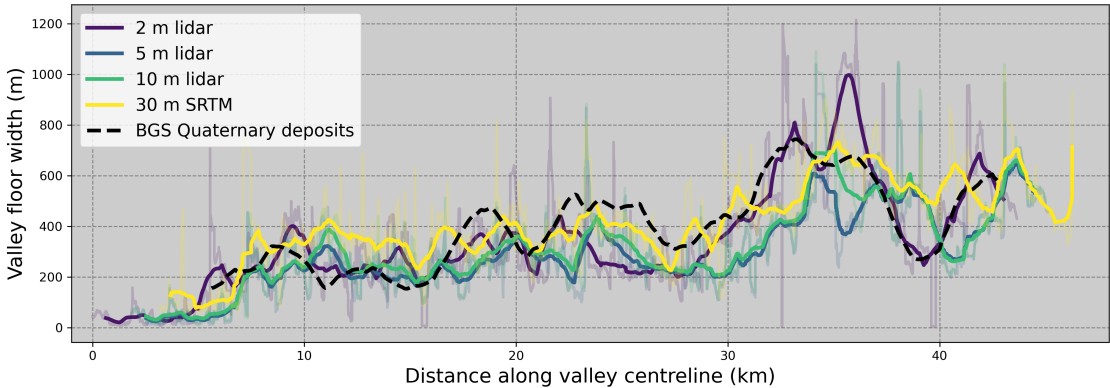

**Figure 7.** Valley floor width along the valley centreline for the Weardale catchment, tested on DEMs of varying resolution. We show the results from lidar-derived datasets at 2 m, 5 m, and 10 m, along with the SRTM 30 m dataset. The black dashed line shows the results from the valleys identified using the BGS Quaternary deposits maps (scale of 1:50,000, corresponding to a DEM resolution of ≈ 25 m).

Table 1 shows the mean valley floor width and standard deviation calculated for Weardale from each DEM resolution as well as from the BGS Quaternary deposits for comparison. We found that the mean valley floor widths tend to increase with the

DEM resolution, with the highest valley floor widths calculated from the SRTM 30 m dataset ($416 \pm 179$ m). This is likely due to the edges of the valley being smoothed with coarsening DEM resolution, as well as the upper portions of the valley which have narrower widths being undetectable from the coarser resolution DEMs. The widths calculated from the SRTM dataset were closest to that of the Quaternary deposit mapping, with a mean width difference of 45 m between the two datasets. The Quaternary deposits are mapped at a scale of 1:50,000, which corresponds to a DEM resolution of ≈ 25 m. This suggests that

the automated technique performs well when compared to independent measurements mapped at a similar resolution.

| Valley extraction method | DEM resolution (m) | Mean valley floor width (m) | Standard deviation (m) |
|---|---|---|---|
| Automated | 2 | 335.57 | 223.59 |
| Automated | 5 | 291.43 | 170.85 |
| Automated | 10 | 313.33 | 179.07 |
| Automated | 30 | 415.56 | 179.47 |
| BGS Quaternary deposits | ≈25 | 390.06 | 194.91 |

**Table 1.** Mean and standard deviation of valley floor widths calculated from Weardale at varying DEM resolutions, and from automated valley floor mapping compared to valley floors mapped from the BGS Quaternary deposits.





### 3.3 Russian River, California, USA

We then tested the method in a downstream reach of the Russian River, Northern California where a 1:24,000 geological map was available. This site allowed the evaluation of the algorithm's effectiveness in a non-glaciated landscape consisting of Quaternary alluvium and fluvial terrace deposits. We followed the same approach as for the BGS Quaternary maps, classifying
alluvium and terrace deposits as 'fluvial' and any other Quaternary sediments as 'non-fluvial' (Figure 8). We then ran the algorithm using the automated valley extraction and by ingesting the USGS Quaternary maps. Figure 8 shows that there was a very good agreement between the widths extracted from the automated and USGS-derived widths, with a mean width of $348 \pm 136$ m estimated from the automated technique and a mean width of $417 \pm 192$ m estimated from the USGS maps. Figure 9 shows the downstream distribution of calculated valley floor widths. The automated widths in this site tended to be
slightly narrower than the BGS-derived widths, with a mean percentage difference of 18% between the two datasets. Close inspection of these maps (see panels to left in Figure 8) suggest that the Quaternary units from the USGS maps include the toes of hillslopes.

## 4  Valley floor width and drainage area

Our method allows the continuous extraction of valley floor width downstream along channels over large spatial scales, pro-
viding an unprecedented opportunity to test theories of valley floor width evolution. To demonstrate this, we performed simple tests of the modelled relationship between valley floor width and drainage area (Section 1.1) over a range of different spatial scales.

### 4.1  Small scale: Cumberland Plateau, Kentucky, USA

Firstly, we focused on small tributary basins in the Cumberland Plateau, Kentucky. The Cumberland Plateau is a dissected
plateau to the west of the Appalachian Mountains and to the south of the contiguous Allegheny Plateau. We focused on this region as we wished to examine controls on valley floor width in a relatively simple region: the Cumberland Plateau is tectonically inactive, has relatively homogeneous geology consisting of shallow dipping Carboniferous sediments, was unglaciated through the Quaternary (Sugden, 1977), and has a humid, temperate climate (Phillips et al., 2010). Our hypothesis is that in this relatively homogeneous landscape, we should find that valley floor width can be well-approximated as a power law of
drainage area following Equation 1, and that the values of $c_v$ and $K_v$ that can be identified from this power law should be reasonably consistent across different channels. To test this hypothesis we focused on ten valleys which are tributaries of the South Fork and Middle Fork Kentucky Rivers, which form part of the Ohio River Basin (Figure 10a). These valleys range in drainage area from approximately 7 - 79 km$^2$. We derived DEMs from the USGS 3DEP data at 1 m resolution for nine of the valleys, and at 2 m resolution for Bullskin Creek due to its larger catchment area. We extracted the valley centrelines and valley
floor widths for each catchment using the same methods as for previous sites. We extracted drainage area along the steepest descent flow trace for each valley and calculated the nearest point along the flow trace for each valley floor width to obtain a





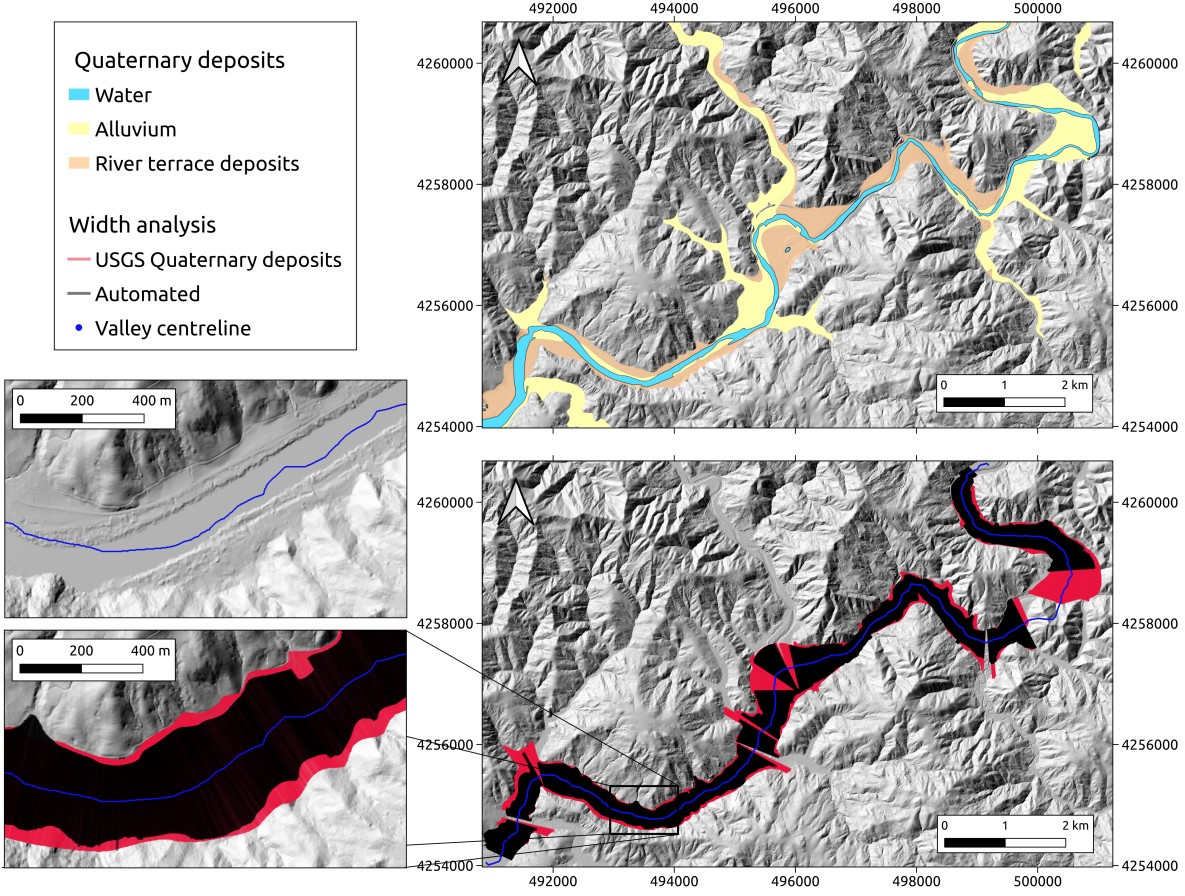

**Figure 8.** Top panel: hillshade of the Russian River showing the extent of superficial Quaternary deposits mapped by the USGS from Knudsen et al. (2000). Bottom panel: valley floor widths extracted using the automated method (black) and those derived from the USGS Quaternary maps (red). The valley centreline is shown in blue. Note the section of the channel to the east of the map (around UTM 500000, 4258000) where the centreline goes across hillslopes: this is due to an isolated hill in the middle of the valley, which obstructs the extraction of an accurate centreline. The zoomed-in map shows the typical width distribution for the site, where the automated widths are generally narrower than those derived from the USGS maps. These show that USGS Quaternary maps tend to include hillslope toes as part of the Quaternary deposits. The coordinate system is WGS84 UTM Zone 10N.

corresponding drainage area for each measurement. Following calculation of the width at 1 m, we filtered the dataset to remove width measurements that were more than 10% wider than the mean width over a moving window of 500 m. This serves to remove widths at tributary junctions where the algorithm generally results in anomalously wide measurements if the floodplain

extends into the tributary.





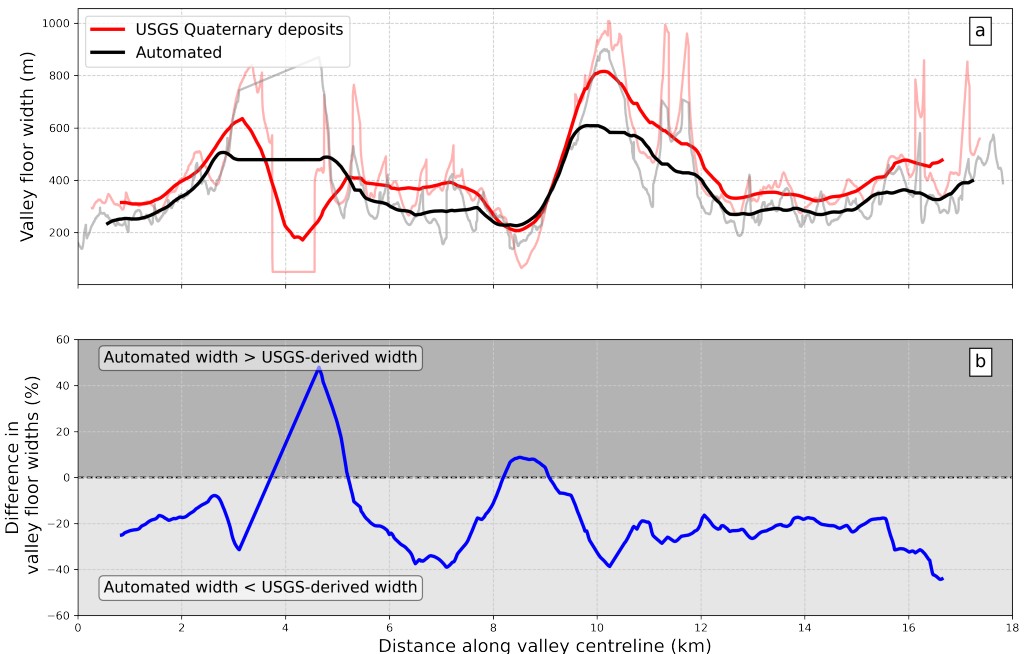

**Figure 9.** Results of valley floor width comparison for the Russian River. (a) Valley floor width from upstream to downstream along the centreline for the automated method (black) and derived from the USGS Quaternary maps (red). The light grey and red lines show the raw data, and the darker lines show a rolling average of width over a 500 m window. (b) Percentage difference between the two datasets along the centreline, where positive values indicate a wider automated width than USGS-derived, and negative values indicated a narrower automated width than USGS-derived.

We fit a power-law relationship between width and area for each basin after Equation 1 and calculated the widening coefficient $K_v$ and the $c_v$ exponent (Table 2). We found that there was generally a positive relationship observed between width and drainage area. In eight of the ten basins a positive $c_v$ value was observed: the mean $c_v$ of these basins was 0.32 with a standard deviation of 0.15. However in two basins, Short Creek and Stinnett Creek, there was no increase in valley floor width with increasing drainage area (Figure 11). For those basins which did show a positive relationship between width and area, the goodness of fit varied between basins from $R^2 = 0.78$ in Flat Creek to $R^2 = 0.26$ for Sugar Creek. Figure 11 shows examples of the relationship between valley floor width and drainage area for selected basins, demonstrating the variability between basins. Plots for all basins can be found in Supplementary Figure 1. We normalised $K_v$ by a reference $c_v$ of 0.21 which was the mean $c_v$ from the ten basins.






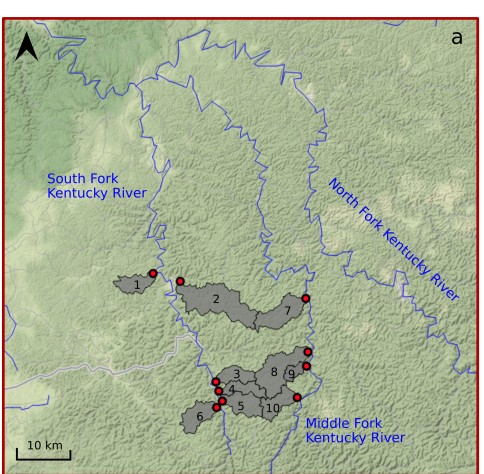

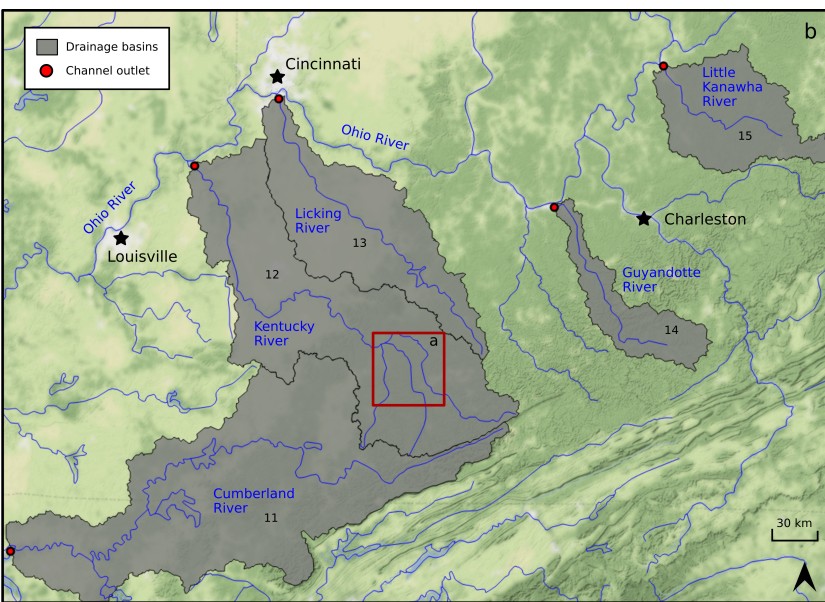

**Figure 10.** Location of the sites for width-area analysis in the Cumberland and Allegheny Plateaus, southwestern Appalachians, USA. a) Small creeks in the Cumberland Plateau; b) large rivers in the Cumberland and Allegheny Plateaus. The red circles mark the outlet of each catchment analysed, labelled 1-15. The tributary where the North and South Forks of the Kentucky River join is located at 37°34'13"N, 83°42'39"W (WGS84). Map tiles by Stamen Design under CC BY 3.0; data © OpenStreetMap contributors. Distributed under the Open Data Commons Open Database License (ODbL) v1.0.

Our results are consistent with the hypothesis that width can be well-approximated as a power law function of drainage area, and that the $c_v$ exponent is reasonably consistent across different channels. However, the lack of valley widening in two of the basins (Short Creek and Stinnett Creek), as well as the variation in the goodness of fit across all basins, suggests that there is heterogeneity in valley widening. Short Creek is, fittingly, the smallest catchment that we analysed, with a drainage area of 7.43 km$^2$. The lack of widening signal in this basin may be a function of the smaller range of drainage areas available to

perform the fit, although we note that Crane Creek is also a small catchment of 15 km$^2$ which shows a clear widening signal. The valley in Stinnett Creek shows a relatively complex pattern of aggradation at the upstream portion of the valley, followed by a narrowing at around 6 - 8 km downstream along the valley centreline. From inspection of satellite imagery, it appears that there has been significant development of mountaintop removal mining in the lower part of the catchment since 1985. The mountaintop removal occurs up Knoblick Branch of Stinnett Creek: wide valleys occur upstream of Knoblick Branch whereas

there is valley narrowing just below the tributary junction. It is possible that enhanced sediment delivery to the channel through Knoblick Branch has caused valley aggradation upstream and altered the natural signal of valley widening in this catchment. Reed and Kite (2020) showed that mountaintop removal mining has caused significant landsliding in central Appalachia: landslide dams and associated sediment delivery is also a potential control on valley floor width (e.g. Korup, 2004; Lancaster, 2008).





**Figure 11.** Width-area analysis for rivers in the Cumberland Plateau, Kentucky, USA. a) Fitted power-law relationships for each of the rivers, showing eight of the rivers demonstrated a positive $W_v$-$A$ relationship while two of the rivers showed no relationship between $W_v$ and $A$. Example width-area plots for b) Flat Creek; c) Short Creek; d) Bullskin Creek; and e) Hell for Certain Creek showing variability between basins.





| Valley name | ID | Drainage area (km$^2$) | $c_v$ | $K_v$ | Range of normalised $K_v$ | $R^2$ |
|---|---|---|---|---|---|---|
| Crane Creek | 1 | 15.06 | 0.24 | 2.09 | 2.55 - 4.27 | 0.71 |
| Bullskin Creek | 2 | 79.06 | 0.3 | 1.34 | 0.93 - 8.6 | 0.55 |
| Sugar Creek | 3 | 44.82 | 0.36 | 0.25 | 0.69 - 7.82 | 0.26 |
| Gilbert's Big Creek | 4 | 44.82 | 0.49 | 0.04 | 1.13 - 15.96 | 0.62 |
| Elisha Creek | 5 | 21.12 | 0.2 | 2.31 | 1.38 - 6.43 | 0.32 |
| Flat Creek | 6 | 43.06 | 0.56 | 0.01 | 1.02 - 6.65 | 0.78 |
| Hell for Certain Creek | 7 | 27.71 | 0.14 | 5.68 | 1.26 - 2.74 | 0.39 |
| Rockhouse Creek | 8 | 39.48 | 0.23 | 1.77 | 1.22 - 5.49 | 0.38 |
| Short Creek | 9 | 7.43 | -0.09 | 192.87 | 1.32 - 3.95 | 0.08 |
| Stinnett Creek | 10 | 18.68 | -0.13 | 305.51 | 0.63 - 3.43 | 0.20 |
| Cumberland River | 11 | 24019.6 | 0.37 | 0.08 | 0.07 - 1.62 | 0.46 |
| Kentucky River | 12 | 16534.8 | 0.33 | 0.14 | 0.06 - 2.12 | 0.37 |
| Licking River | 13 | 9599.67 | 0.22 | 2.71 | 0.1 - 2.65 | 0.19 |
| Guyandotte River | 14 | 3364.67 | 0.26 | 0.69 | 0.1 - 1.85 | 0.34 |
| Little Kanawha River | 15 | 5828.98 | 0.34 | 0.16 | 0.1 - 2.12 | 0.51 |

**Table 2.** Width-area analysis for rivers in the Cumberland and Allegheny Plateaus, Kentucky and West Virginia, USA. The ID of the valley correlates to the locations in Figure 10. $c_v$ and $K_v$ are calculated from power law fits to width and drainage area along each channel. Normalised $K_v$ is calculated using a reference value of $c_v = 0.21$ for valleys 1 - 10 and $c_v = 0.3$ for valleys 11 - 15.

### 4.2 Scaling up: the southwestern Appalachians, USA

Following on from this high resolution, small scale analysis, we extracted valley floor width and drainage area from large rivers draining the Cumberland and Allegheny Plateaus in Kentucky and West Virginia to test whether larger catchments show the same trends of valley floor width and drainage area. For simplicity we will refer to these regions together as the Appalachian Plateau. We analysed five major rivers which drain into the Ohio River basin: the Cumberland River, Kentucky River, Licking River, Guyandotte River, and Little Kanawha River (Figure 10b). These channels drain the same tectonically inactive southern Appalachians as the smaller creeks, with the same underlying geology of dipping Carboniferous sandstones, siltstones and shales. However, their large extent means that they may drain across more units of varying hardness. We extracted valley floor width and drainage area from the 30 m SRTM dataset downloaded from OpenTopography, following the same approach as for the small creeks. Due to the coarser resolution, extracting a valley centreline was not necessary as the D8 flow routed channel was already in the centre of the valleys. Figure 12 shows the width-area results for these large rivers. The relationship between $W_v$ and $A$ is remarkably consistent across these channels, with a mean $c_v$ of 0.3 and a standard deviation of 0.06. This result shows that our method is successful at identifying meaningful valley floor widths across large spatial scales (the largest basin is 24,019 km$^2$), and that it is applicable on 30 m SRTM data available at a near-global scale. Furthermore, we conclude that





in a tectonically inactive, non-glaciated region with relatively homogeneous lithology, reliable $c_v$ values can be extracted from
topographic data.

Different $c_v$ values have been linked to different valley widening mechanisms, such as the undercutting-slump and total
block erosion models put forward by Langston and Tucker (2018). Our mean $c_v$ value of 0.3 for the Appalachian Plateau is
similar to the values reported for intermediate hardness lithologies, mostly consisting of marly limestones, by Langston and
Temme (2019). Their results of numerical modelling runs simulating end-member valley erosion mechanisms found $c_v \approx 0.25$
for a resistant valley widening mechanism, where the entire valley wall height must be removed before widening can occur;
while $c_v \approx 0.4$ for an erodible mechanism where undercutting and slumping can occur. Our results for the Appalachian Plateau
fall between these two end-members. This suggests that valley widening in the Appalachian Plateau takes place through a
combination of these different end-member scenarios. Additional research into the size of material eroded from valley walls
in the region and the grain size of sediment within the active channel and floodplain would help to constrain different valley
widening mechanisms. Furthermore, a $c_v$ value of 0.3 is also similar to the analysis of Schanz and Montgomery (2016) for
marine sedimentary units in Washington State ($c_v$ = 0.34) and Brocard and van der Beek (2006) for the western Alps, which
found that $c_v$ generally fell between 0.3 and 0.4. Brocard and van der Beek (2006) found more variability in their normalised
$K_v$ values, which is unsurprising considering the lithological and tectonic variability present within the western Alps compared
to the Appalachian Plateau.

Gallen (2018) constrained the vertical fluvial erodibility parameter, $K_d$, from across the Appalachians, and found that the
Carboniferous sandstones, shales and siltstones of the Appalachian Plateau had intermediate $K_d$ values of $\approx$ 1.0 - 1.5 x
$10^{-6}$ m$^{0.1}$ yr$^{-1}$, compared to the more resistant rocks of the Blue Ridge ($K_d \approx$ 0.5 x $10^{-6}$ m$^{0.1}$ yr$^{-1}$) or more erodible
carbonates, shales and siltstones of the Valley and Ridge province ($K_d \approx$ 1.6 - 2.0 x $10^{-6}$ m$^{0.1}$ yr$^{-1}$). Our estimate of the
normalised lateral erodibility parameter, $K_v$, ranges from 0.07 - 2.65 m/m$^2$ for the Appalachian Plateau. Absolute values are
not readily comparable because of differences in units between the two parameters, but, importantly, we find that the region has
intermediate lateral erodibility values compared to other studies, as well as the intermediate vertical erodibility values reported
by Gallen (2018).

## 5    Beyond single channels: valley floor width as a network-scale metric

In this contribution we have demonstrated continuous measurements of valley floor width downstream along channels and
across landscapes. This approach provides new opportunities to explore how valley floor width changes between basins at
an unprecedented spatial scale: here we have shown how these data can be used to test current models of valley floor width
evolution by exploring the relationship between valley floor width and drainage area. However, we are not limited to exploring
only main stem channels as we have presented in this contribution: this technique can be applied to the entire fluvial network
extracted from a DEM. This means we can explore how valley floor width varies across scales within a network, and potentially
use valley floor width as an orogen-scale topographic metric similar to normalised channel steepness, for example. Channel
steepness is a widely used metric in the geomorphic community because i) it can easily be calculated from DEMs using only





**Figure 12.** Width-area analysis for large rivers in the Appalachian Plateau, Kentucky and West Virginia, USA. Example width-area plots are shown for each river along with a compiled plot of the fitted power laws for each river in the top left.

Earth **Surface**
**Dynamics**
Discussions

elevation and drainage area; and ii) it has been shown to correlate well with inferred uplift rates (e.g. Snyder et al., 2000; Kirby and Whipple, 2001; Wobus et al., 2006). However, reliance on channel steepness alone neglects the lateral component of fluvial adjustment to external forcing, and only considers network change in the vertical domain. Using valley floor width
in combination with channel steepness would provide the opportunity to explore both vertical and lateral fluvial signatures of climate, tectonics, lithology, or drainage reorganisation. Here we have shown that continuous measurements of downstream valley floor width can be reliably extracted from 30 m digital elevation models which are available at a near-global scale. Future research could explore how valley floor width changes across uplift gradients, how structures such as fold-and-thrust belts affect lateral channel erosion, or investigate the impact of changing sediment supply or characteristics on valley evolution.

*Code and data availability.* The code for valley floor width extraction is available as part of the LSDTopoTools software package: https: //github.com/LSDtopotools/lsdtt_opencv_docker. A tagged version of the LSDTopoTools software is available on Zenodo (Mudd et al., 2021). All parameter files used for valley extraction are available as part of the supplement.

*Author contributions.* FJC and SMM developed the valley floor width extraction algorithm; FJC and EFW downloaded the topographic datasets and ran the analyses. FJC wrote the manuscript and created the figures with contributions from all authors.

*Competing interests.* The authors declare that no competing interests are present.

*Acknowledgements.* This work was supported by a Research Development Fund grant from Durham University. LSDTopoTools code development for this contribution was supported by NERC grants NE/P012922/1 and NE/S009000/1. We would like to thank OpenTopography for providing access to the USGS 3DEP data, and Steffi Tofelde and Andy Wickert for fruitful discussions.





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
