# Peer review of "Continuous measurements of valley floor width in mountainous landscapes"

_Earth Surface Dynamics, 2022_

## Referee Comment (RC2)

**Peer Review**: Continuous Measurements of valley floor width in mountainous landscapes

Authors: Clubb, F.J., Weir, E.F., Mudd, S.M.

Journal Earth Surface Dynamics

Review by: Matthew Morriss, PhD (USGS)

**Overall impression**

In this paper, the authors lay out a method for continuously measuring valley width along an individual channel or within a network. This paper highlights an important hole in our measurements of channel and valley processes; we as geomorphologists have numerous methods for extracting continuous or nearly continuous data on the vertical components of channels, their $\Delta Z$, but not the changes in channel or valley widths $\Delta W$. The paper lays out the clear implications of changes in valley width and the studies that have previously examined this measurement ($W_v$). The authors have a clear grasp of the literature on both tectonic and fluvial geomorphology and where their study can add to the understanding of valley widening processes. The authors test their methods in a series of tectonically quiescent regions with both homo- and heterogeneous rock types to evaluate the potential scaling effects of drainage area with valley width proposed by other authors (equation 1; e.g. Tomkin et al., 2003; Brocard and van der Beek, 2006; Schanz and Montgomery, 2016).

$$W_v = K_v A^{c_v} \qquad\qquad 1$$

Where $K_v$ is the influence of rock type on valley width; $A$ is drainage area, and $c_v$ is a descriptor of how valley width changes with drainage area (positively or negatively).

Based on my critical review of the submitted manuscript, I would recommend this paper be: **Accepted with Revisions.**

I would also like to add that I'm excited for where this paper ends. Specifically, how the applied methods could be incorporated into future large scale, regional analyses of rivers in just the same way as $\chi$ or $k_{sn}$ are often applied around the world today.

- Matthew Morriss, PhD USGS

**Major Comments**

While I am certain this paper is worthy of being accepted, I want to provide the authors with a critical list of comments to better their manuscript moving forward. These will largely be comments line by line through the text. I will also provide a second set of comments for minor issues I noticed in my review. I will also try to preserve the tongue-in-cheek language the authors spread throughout their paper, for the benefit of subsequent readers.

Line 57: The authors base the power-law relationship for valley width and drainage area on the firm footing of literature; however, here, it is revealed that empirical studies suggested this power-law relationship have been conducted on bedrock rivers. I have not found another reference or description through the text of the test cases examined by the authors as bedrock rivers. It would further strengthen this paper to justify that the test sites are in fact bedrock channels and not alluvial channels unless the authors seek to make a more general case about valley width scaling beyond just bedrock rivers. A potentially germane reference for alluvial rivers is Leopold and Maddock (1953) or some of the subsequent citing literature.

**Figure 1:** This figure does a good job of summarizing the preceding sections forcings on channel width. However, the figure itself does not match all the forcings described in the inset text. Perhaps worth adding "Uplift" and "Incision" as arrows to the figure otherwise it seems disconnected with the text directly above.

Lines 151-158: Again, are these bedrock channels? That seems key to your later investigations of power-law scaling.

Line 165-167: This will not be my last comment regarding several places where methods are mentioned but not documented nor shown to the reader. Please remember to "show and tell" your methods. I – as the reader – cannot conjure up the numbers you used and to protect your work going forward its important to add this type of detail. "Thresholds for elevation above the channel and local slope can either be calculated statistically using qq plots of the distribution of slope and elevation across the landscape, or they can be set manually by the user." This text reads more like a manual for code rather than a paper. To improve it, I would recommend adding a table / figure to the supplement highlighting this workflow and examples of the thresholds. You needn't do it for all of your sites, but at least once please.

Line 175-180: I am unclear if the process of using either the steepest descent path or the method highlighted in the next few paragraphs is an automated choice and if so how it is made or if the user interacts at that step?

Line 181-203: This section of text describing the method for delineating the valley center line, I found to be exceedingly difficult to follow. To clarify this section, I recommend the following:

- Consider numbering the steps in the process. The text, as written, uses the phrase "We begin" or "We then" or "we X" at least six times in this section. Could replace and make easier to follow with a numbering system.
- I think this method is the perfect place for a diagrammatic figure. I realize the authors cite figure 2, but that figure doesn't show the nuts and bolts for how the code is working. Below are a few examples pulled from google images of what I'm imagining:

[Figure]

- This 👆 is obviously not a perfect example as its not made for your paper, but something diagrammatic like this figure or to that effect would vastly clarify 1) the workflow being conducted here and 2) the decision points being made either automatically or by someone interacting with the code. Rather than having to draw out an example for myself as the reviewer, or reader, provide your own visualization as there are a lot of concepts being discussed that a good visual aid would help.

Line 188-189: The method describes subtracting elevation by a multiplying the distance from the bank with a "scaling factor." What is this "Scaling factor"? Again, please opt toward explicit discussions of such variables and highlight what they are and how you chose them.

Line 191-192: How many times is the valley carved and filled? The author doesn't describe or detail that in the text. Is it provided as an option for the users? Or is it in Linsday (2016?)

**Figure 2.** This holds true for all maps in this paper, if you make a map, it needs a scale bar, a north arrow, and a legend. I will call out each figure individually in my comments, but I want to see the authors add at least a scale bar and north arrow to all their maps. Yes, I know that the projection is UTM so the coordinates on the margins are meters, but those labels are quite small; a scale bar could still be useful! I also can't tell the difference between the "dark blue" and "light blue" the authors cite in the caption – perhaps try different, higher-contrast colors?

Line 229: The River Tweed is gravel bedded? Does that mean your assumption of a power-law is valid? Or perhaps the power-law relationship extends to non-bedrock rivers?

Line 275: The authors make the argument that mean valley floor widths tend to increase with increasing DEM grid size; however, the $3^{rd}$ widest mean valley floor width is for the 2 meter data. Can this be explained? Why the 5 meter and 10 meter data seem anomalously successful in their performance compared to the 2 meter data? Also, the 2 meter data has the largest variance of any dataset, but perhaps that's to be expected?

Line 370-377: It would behoove the authors to add in another reference to the actual values of $K_v$ determined from other studies.

This Study: $K_v$ = 0.07-2.65 m m$^{-2}$

(Brocard and van der Beek, 2006): 8-160 m km$^{-0.8}$

(Langston and Tucker, 2018): 0.16 +/- 0.052 m m$^{-2}$ (model result)

(Tomkin et al., 2003): 2.81 m km$^{-0.8}$

These values above are from the papers cited in the introduction. Note that the units here also do not match the units used by the authors. This seems like a great place for a table restating the $K_v$ values from the literature, with matching units to the authors. This will add power to the statement regarding this study finding intermediate range of values for $K_v$. As it is written, the audience must flip back to the Introduction to confirm their intermediate status and then check that the units match, which they don't. Confirm similar units, then confirm intermediate erodibility values in this study as compared with others.

**Minor Comments; Line by line Edits**

Line 12: How well does your method extract valley width? Good place to put some #s in the abstract

Line 43: Another reference is needed here: (Wang and Willett, 2021).

Line 104-15: Fantastic explanation – very much enjoyed this.

Line 113-114: Perhaps an exception to the lack of post-glacial models is (Leith et al., 2018). Could add this citation and that there are "limited studies on post-glacial fluvial erosion."

Line 233: "of 5 m." to "We found …" is an abrupt transition consider rewording.

Line 235: in this width calculation and henceforth in all other width calculations is it the $width$ $+/- 1 \sigma$? Or $2\sigma$? Just would appreciate your clarifying and then its easier to know for all other measurements.

**Figure 3.** North arrow and scale for two main maps and inset map please.

**Figure 4b**. These plots could benefit the reader with a second y axis that is the actual difference between the two datasets in meters. Shouldn't be too difficult to do.

**Figure 5.** Nice work on the scale for insets. Please label each panel for easy comparison to caption and add scale and north arrow to main map.

Line 290: Authors reference "BGS;" however this section is for the Russian River in California. I think they mean: USGS.

**Figure 8**: 1) Please label all map plate pieces and insets and refer to them in your caption.  2) Add north arrows to all main maps. Thank you for including a scale bar! Labeling insets may allow you to remove the tie-lines between one inset and main figure.

Line 320: Short Creek and Stinnett Creek don't display only the lack of increasing valley width with drainage area; they show a negative relationship! Maybe be a bit more clear about this in this sentence.

**Figure 10**: 1) Add state lines to reference map. 2) potentially switch a and b to have smaller scale map first and then larger scale map. 3) In caption, please use decimal degrees not DMS – easier for reviewing audience to look at locations used in paper.

Line 374: I believe standard notation should read m m$^{-2}$.

Line 391: Here[,] ← add the comma after "Here"

**Supplementary Material:**

This reviewer had no comments on the supplementary material attached with this manuscript. However, I did try to find the code associated with this project and only found the link at the end of the text lead to a github page for downloading all of LSD Topotools. Is there a direct link to the code for the tool described in this paper that the authors can provide? Perhaps the direct link to the Zenodo? https://zenodo.org/record/5788576#.Yi9eCnrMIuU

**References**

Brocard, G.Y., and van der Beek, P.A., 2006, Influence of incision rate, rock strength, and bedload supply on bedrock river gradients and valley-flat widths: Field-based evidence and calibrations from western Alpine rivers (southeast France), *in* Tectonics, Climate, and Landscape Evolution, Geological Society of America, doi:10.1130/2006.2398(07).

Langston, A.L., and Tucker, G.E., 2018, Developing and exploring a theory for the lateral erosion of bedrock channels for use in landscape evolution models: Earth Surface Dynamics, v. 6, p. 1–27, doi:10.5194/esurf-6-1-2018.

Leith, K., Fox, M., and Moore, J.R., 2018, Signatures of Late Pleistocene fluvial incision in an Alpine landscape: Earth and Planetary Science Letters, v. 483, p. 13–28, doi:10.1016/j.epsl.2017.11.050.

Schanz, S.A., and Montgomery, D.R., 2016, Lithologic controls on valley width and strath terrace formation: Geomorphology, v. 258, p. 58–68, doi:10.1016/j.geomorph.2016.01.015.

Tomkin, J.H., Brandon, M.T., Pazzaglia, F.J., Barbour, J.R., and Willett, S.D., 2003, Quantitative testing of bedrock incision models for the Clearwater River, NW Washington State: QUANTITATIVE TESTING OF RIVER INCISION MODELS: Journal of Geophysical Research: Solid Earth, v. 108, doi:10.1029/2001JB000862.

Wang, Y., and Willett, S.D., 2021, Escarpment retreat rates derived from detrital cosmogenic nuclide concentrations: Earth Surface Dynamics, v. 9, p. 1301–1322, doi:10.5194/esurf-9-1301-2021.

---

## Author Response (AR1)

**Authors' Response to Reviews of**

**"Continuous measurements of valley floor width in mountainous landscapes"**

Fiona J. Clubb, Eliot F. Weir, Simon M. Mudd
*ESURF,*
* * *
**RC:** *Reviewers' Comment*,     AR: Authors' Response,     ☐ Manuscript Text

**1. Reviewer #1 Comments**

**RC:** *393 or explore what drives the variability in valley width! Can't help noticing the order of magnitude variability in width over miniscule changes in drainage area (fig 12)!*

**AR:** Thank you for your positive review and support of our manuscript. This is a good point. We have added a mention of this in the text as suggested.

> Future research could explore the drivers of valley floor width variability across landscapes, how width changes across uplift gradients, how structures such as fold-and-thrust belts affect lateral channel erosion, or investigate the impact of changing sediment supply or characteristics on valley evolution.

**RC:** *Fig 7: probably should add the note about line colors and rolling averages you have in the other figure captions in case someone jumps to this figure.*

**AR:** Thanks. We have done this.

**2. Reviewer #2 Comments**

**RC:** *While I am certain this paper is worthy of being accepted, I want to provide the authors with a critical list of comments to better their manuscript moving forward. These will largely be comments line by line through the text. I will also provide a second set of comments for minor issues I noticed in my review. I will also try to preserve the tongue-in-cheek language the authors spread throughout their paper, for the benefit of subsequent readers.*

**AR:** Thank you very much for your support of our manuscript and your comments. We have answered each comment in the responses below.

**RC:** *Line 57: The authors base the power-law relationship for valley width and drainage area on the firm footing of literature; however, here, it is revealed that empirical studies suggested this power-law relationship have been conducted on bedrock rivers. I have not found another reference or description through the text of the test cases examined by the authors as bedrock rivers. It would further strengthen this paper to justify that the test sites are in fact bedrock channels and not alluvial channels unless the authors seek to make a more general case about valley width scaling beyond just bedrock rivers. A potentially germane reference for alluvial rivers is Leopold and Maddock (1953) or some of the subsequent citing literature.*

AR: We now say the following in the text at this point:

> The use of the term "bedrock systems" requires some explanation. In studies using equation 1 authors did not necessarily look for a strath terrace: the valley floor away from the channel could be alluviated. However a valley flowing though hills must, at some point, have incised, so bedrock incision is inferred even if the valley contains alluvial sediment.

Leopold and Maddock (1953) is not relevant here: they do not explore valley width and our aim is not to comment on the extensive literature on channel width.

RC: *Figure 1: This figure does a good job of summarizing the preceding sections forcings on channel width. However, the figure itself does not match all the forcings described in the inset text. Perhaps worth adding "Uplift" and "Incision" as arrows to the figure otherwise it seems disconnected with the text directly above.*

AR: Done.

RC: *Lines 151-158: Again, are these bedrock channels? That seems key to your later investigations of power-law scaling.*

AR: We have added the following text for clarity:

> The method extracts the distance between valley walls (i.e., from hillslope to hillslope) where a valley floor is present, which could include either bedrock or alluvial channel reaches.

RC: *Line 165-167: This will not be my last comment regarding several places where methods are mentioned but not documented nor shown to the reader. Please remember to "show and tell" your methods. I – as the reader – cannot conjure up the numbers you used and to protect your work going forward its important to add this type of detail. "Thresholds for elevation above the channel and local slope can either be calculated statistically using qq plots of the distribution of slope and elevation across the landscape, or they can be set manually by the user." This text reads more like a manual for code rather than a paper. To improve it, I would recommend adding a table / figure to the supplement highlighting this workflow and examples of the thresholds. You needn't do it for all of your sites, but at least once please.*

AR: We have added a table to the Supplementary Information (Table S1) which documents the slope and relief thresholds used for each field site in the paper.

RC: *Line 175-180: I am unclear if the process of using either the steepest descent path or the method highlighted in the next few paragraphs is an automated choice and if so how it is made or if the user interacts at that step?*

AR: This is a choice that can be made by the user. We have clarified this in the text:

> We therefore provide an option of extracting the valley centreline from which to determine valley floor width (Figure 2). This option can be chosen by the user after inspection of DEM and the initial results of the floodplain extraction.

RC: *Line 181-203: This section of text describing the method for delineating the valley center line, I found to be exceedingly difficult to follow. To clarify this section, I recommend the following: Consider numbering the steps in the process. The text, as written, uses the phrase "We begin" or "We then" or "we X" at least six times in this section. Could replace and make easier to follow with a numbering system. I think this*

*method is the perfect place for a diagrammatic figure. I realize the authors cite figure 2, but that figure doesn't show the nuts and bolts for how the code is working. Below are a few examples pulled from google images of what I'm imagining [image omitted]: This is obviously not a perfect example as its not made for your paper, but something diagrammatic like this figure or to that effect would vastly clarify 1) the workflow being conducted here and 2) the decision points being made either automatically or by someone interacting with the code. Rather than having to draw out an example for myself as the reviewer, or reader, provide your own visualization as there are a lot of concepts being discussed that a good visual aid would help.*

AR: We have rewritten the explanation of the centreline extraction to make it clearer, and have added in an equation to explain the scaling factor in response to the comment below. We have also added a figure to illustrate the computations involved.

> Firstly, we take the valley floor mask and calculate the distance to the nearest valley wall for each pixel. We record i) the elevation of the nearest valley wall pixel; and ii) the distance to the valley wall. Using a moving window with a radius roughly equal to the valley width, we set the elevation of each valley floor pixel to the minimum elevation of the valley wall within the window, $z_{min}$. This creates an initial smooth valley surface which is at the minimum elevation of the valley walls. We then artificially decrease the elevation of this surface to create a new elevation for each pixel, $z_t$, weighting the decrease in elevation by distance away from the valley wall, $d_w$, such that:
>
> $$z_t = z_{min} - \alpha d_w. \tag{1}$$
>
> The effect of this is to form a V-shaped valley trough with the deepest part of the valley occurring in the middle (Figure 1). $\alpha$ is a scaling factor that determines the magnitude of the elevation decrease. In our analysis we set $\alpha = 0.5$, but this can be modified by the user to increase or decrease the slope of the V-shaped trough.

RC: *Line 188-189: The method describes subtracting elevation by a multiplying the distance from the bank with a "scaling factor." What is this "Scaling factor"? Again, please opt toward explicit discussions of such variables and highlight what they are and how you chose them.*

AR: We have modified the text (see response to comment just above) to explain this scaling factor and how we set this in our analysis.

RC: *Line 191-192: How many times is the valley carved and filled? The author doesn't describe ordetail that in the text. Is it provided as an option for the users? Or is it in Linsday (2016?)*

AR: We have modified the text after these lines to explain this further:

> The number of carving and filling iterations can be set by the user: based on our testing we find that 5 iterations gives a balance between an accurate centreline and time required for computation, and we use this value for subsequent analysis.

RC: *Figure 2. This holds true for all maps in this paper, if you make a map, it needs a scale bar, a north arrow, and a legend. I will call out each figure individually in my comments, but I want to see the authors add at least a scale bar and north arrow to all their maps. Yes, I know that the projection is UTM so the coordinates on the margins are meters, but those labels are quite small; a scale bar could still be useful!*

[Figure]

Figure 1: Step by step automated extraction of the channel centreline from topographic data: example from 1m lidar DTM from Gabilan Mesa, California. Coordinates are WGS84 UTM Zone 10N. The floodplain mask is first extracted using the method of Clubb et al. (2017). (1) The next step calculates the distance of every pixel within the floodplain to the nearest non-floodplain pixel (or "valley wall" pixel). (2) In addition, the minimum valley wall pixel elevation is mapped onto every pixel in the floodplain; the minimum elevation is determined by the lowest valley wall elevation within a circular window around each floodplain pixel. (3) An elevation is subtracted from the minimum elevation using equation 1 which incorporates the distance to the nearest valley wall. This results in a "trough" that is roughly V-shaped in cross section. Flow is routed down this trough to extract the valley centreline.

*I also can't tell the difference between the "dark blue" and "light blue" the authors cite in the caption – perhaps try different, higher-contrast colors?*

AR: All maps already have a north arrow. One of the advantages of using UTM is that the axes are in units of length and the axis are explicitly northing and easting. We have added these axis labels. A scale bar in this context follows the definition of chart junk (https://en.wikipedia.org/wiki/Chartjunk) so we have not added these features.

RC: *Line 229: The River Tweed is gravel bedded? Does that mean your assumption of a power-law is valid? Or perhaps the power-law relationship extends to non-bedrock rivers?*

AR: We have clarified in response to previous comments that the algorithm will work regardless of whether the channel is alluvial or bedrock. The river has a few bedrock steps but is mostly gravel bedded. The main feature of this river is its well developed valley floor that sits between hills.

RC: *Line 275: The authors make the argument that mean valley floor widths tend to increase with increasing DEM grid size; however, the 3rd widest mean valley floor width is for the 2 meter data. Can this be explained? Why the 5 meter and 10 meter data seem anomalously successful in their performance compared to the 2 meter data? Also, the 2 meter data has the largest variance of any dataset, but perhaps that's to be expected?*

AR: The 2 m dataset is influenced by a region of wide valley widths at $\approx$ 35 km downstream (manuscript Figure 7) which are not picked up by any of the other datasets. This leads to a higher mean valley width as well as greater variance. Inspection of the hillshade (see Figure 2) shows that this is because there is a low-lying flat region to the north of the valley at this distance downstream: this is identified as valley by the 2 m dataset but not by the 5 m. This variation between DEMs is probably due to subtle differences in the slope and elevation distributions across the landscape and is impossible to get around when dealing with real data. Increased variation in valley width measurements at higher resolution is to be expected as more landscape roughness will be preserved compared to coarser resolutions.

We have already discussed this in our manuscript:

> For example, at $\approx$ 36 km along the valley centreline, the 2 m dataset shows widths of up to 1 km, whereas the 10 m, 30 m, and BGS Quaternary deposits indicate valley floor widths of between 500 - 700 m. Furthermore, the 2 m lidar is able to calculate valley floor widths at the furthest upstream part of the valley, whereas datasets with coarsening DEM resolution can only identify the valleys further downstream. For example, the SRTM dataset only starts identifying valley floor widths at $\approx$ 4 km downstream along the valley centreline.

RC: *Line 370-377: It would behoove the authors to add in another reference to the actual values of Kv determined from other studies. This Study: Kv = 0.07-2.65 m m-2 (Brocard and van der Beek, 2006): 8-160 m km-0.8 (Langston and Tucker, 2018): 0.16 +/- 0.052 m m-2 (model result) (Tomkin et al., 2003): 2.81 m km-0.8 These values above are from the papers cited in the introduction. Note that the units here also do not match the units used by the authors. This seems like a great place for a table restating the Kv values from the literature, with matching units to the authors. This will add power to the statement regarding this study finding intermediate range of values for Kv. As it is written, the audience must flip back to the Introduction to confirm their intermediate status and then check that the units match, which they don't. Confirm similar units, then confirm intermediate erodibility values in this study as compared with others.*

AR: The reviewer is correct that the units of $K_v$ vary. This is because the units are dependent on the value of $c_v$ that is used. In order to compare our $c_v$ values to each other, we calculate normalised $K_v$ with a reference $c_v$

[Figure]

Figure 2: Hillshade showing anomalies between 2 m and 5 m width measurements for Weardale. The 2 m dataset picks up a flat surface to the north of the valley which is not identified by coarser resolution datasets. The bottom panel shows the hillshade and the valley centreline.

value of 0.21 for the Cumberland Plateau small rivers, and 0.3 for the large rivers from the Allegheny Plateau. This difference in units makes it difficult to compare between them, which we state explicitly in line 375. The current comparisons we make with the literature cited by the reviewer are of the $c_v$ values found, not the $K_v$ values.

For the units, we added text to the table caption:

> The units of $K_v$ vary depending on the value of $c_v$: they are m km$^{-2c_v}$. The drainage area is reported in km$^2$.

We have revised the text to clarify that we are comparing $c_v$ values and that we cannot compare $K_v$ values to the literature because of the unit difference:

> Absolute values are not readily comparable because of differences in units between $K_d$ and $K_v$, as well as between $K_v$ values calculated with different values of $c_v$, but importantly we find that the Appalachian Plateau has intermediate $c_v$ exponents compared to other studies, as well as the intermediate vertical erodibility values reported by Gallen (2018).

**RC:**   *Line 12: How well does your method extract valley width? Good place to put some s in the abstract*

 AR:   Done:

> We find that our method extracts similar downstream patterns of valley floor width to the independent datasets in each site, with a mean width difference of 17 - 69 m.

**RC:**   *Line 43: Another reference is needed here: (Wang and Willett, 2021).*

 AR:   This paper is about escarpment retreat and not really relevant. It only has channel backwearing due to tectonic advection but there is no lateral migration in their model. We have added a more relevant paper by Baynes et al. 2018.

**RC:**   *Line 104-15: Fantastic explanation – very much enjoyed this.*

 AR:   Thanks.

**RC:**   *Line 113-114: Perhaps an exception to the lack of post-glacial models is (Leith et al., 2018). Could add this citation and that there are "limited studies on post-glacial fluvial erosion."*

 AR:   Thank you - we have added this reference in.

> Leith et al. (2018) studied valley long- and cross-profiles in the Alpine Rhone River region, Switzerland, and found that valleys typically had a complex pattern of valley floor widths with a mix of narrow bedrock reaches and broad alluviated regions. They suggested that this pattern resulted from glacial sediment deposition as well as over-steepening of valley sides in the downstream direction, causing frequent landslides or debris flows and narrow, constricted valleys. Few models of fluvial erosion and lateral migration exist that have been developed with high-latitude, post-glacial systems in mind, despite their prevalence over large regions of the Earth's surface.

**RC:**   *Line 233: "of 5 m." to "We found ..." is an abrupt transition consider rewording.*

 AR:   We now say:

> We then compared these automatically extracted widths with those extracted using Quaternary maps (Figure 4).

**RC:** *Line 235: in this width calculation and henceforth in all other width calculations is it the +/ 1 ? Or 2? Just would appreciate your clarifying and then its easier to know for all other measurements.*

AR: The error is $1\sigma$. We have clarified this in the text.

**RC:** *Figure 3. North arrow and scale for two main maps and inset map please.*

AR: See response to earlier comment.

**RC:** *Figure 4b. These plots could benefit the reader with a second y axis that is the actual difference between the two datasets in meters. Shouldn't be too difficult to do.*

AR: This information is already shown in panel A which shows the absolute valley widths for each method - the difference can be read from this panel. We prefer not to add this to panel B to keep it clean and easy to understand.

**RC:** *Figure 5. Nice work on the scale for insets. Please label each panel for easy comparison to caption and add scale and north arrow to main map.*

AR: We have labelled the panels. There is already a north arrow. We have added axis labels for northing and easting which are used for the scale on the main map.

**RC:** *Line 290: Authors reference "BGS;" however this section is for the Russian River in California. I think they mean: USGS.*

AR: Fixed.

**RC:** *Figure 8: 1) Please label all map plate pieces and insets and refer to them in your caption. 2) Add north arrows to all main maps. Thank you for including a scale bar! Labeling insets may allow you to remove the tie-lines between one inset and main figure.*

AR: We have labelled each panel. North arrows were already present.

**RC:** *Line 320: Short Creek and Stinnett Creek don't display only the lack of increasing valley width with drainage area; they show a negative relationship! Maybe be a bit more clear about this in this sentence.*

AR: We now say:

> However in two basins, Short Creek and Stinnett Creek, there was an inverse relationship between valley floor width and drainage area, although these two basins had two out of the three lowest $R^2$ values for this relationship amongst all valleys.

**RC:** *Figure 10: 1) Add state lines to reference map. 2) potentially switch a and b to have smaller scale map first and then larger scale map. 3) In caption, please use decimal degrees not DMS – easier for reviewing audience to look at locations used in paper.*

AR: Done

**RC:** *Line 374: I believe standard notation should read m m-2.*

AR: Done

**RC:** *Line 391: Here[,] add the comma after "Here"*

AR: We do not agree that a comma should be placed here.

**RC:** *Supplementary Material: This reviewer had no comments on the supplementary material attached with this manuscript. However, I did try to find the code associated with this project and only found the link at the end of the text lead to a github page for downloading all of LSD Topotools. Is there a direct link to the code for the tool described in this paper that the authors can provide? Perhaps the direct link to the Zenodo? https://zenodo.org/record/5788576.Yi9eCnrMIuU*

AR: The code for this paper is part of the LSDTopoTools main distribution, so we cannot provide a direct link to the valley width code itself. Users should install LSDTopoTools and use the tool provided as part of this distribution. To clarify we added this comment in the code and data availablility section:

> All figures and analyses in this paper were performed using the 'lsdtt-valley-metrics' program that is part of the LSDTopoTools software package version 0.5.

---

## Author Response (AR2)

**Authors' Response to Reviews of**

**"Continuous measurements of valley floor width in mountainous landscapes"**

Fiona J. Clubb, Eliot F. Weir, Simon M. Mudd
*ESURF,*
* * *
**RC:** *Reviewers' Comment*,     AR: Authors' Response,     ☐ Manuscript Text

Dear Wolfgang,

Thank you very much for your careful handling of our paper and your comments. We have responded to each comment below and changed the manuscript accordingly.

Best wishes,

Fiona, Eliot and Simon

**1. Associate Editor Comments**

**RC:** *I am quite pleased to see a paper that deviates from the standard heading-scheme: Introduction, Methods, etc. , although this makes it a bit harder to quickly navigate the manuscript at first sight. However, in the guidelines (https://www.earth-surface-dynamics.net/submission.htmlmanuscriptcomposition) we ask our authors to provide sections titled Introduction and Conclusions. While you have the former, I need to ask you to rename the last section to Conclusions.*

 AR:  We have renamed this section to Conclusions.

**RC:** *369: Really, the D8 flow routed channel was already and always in the valley centre? This is actually hard to believe looking at e.g. the Cumberland River and how it freely meanders in a relatively wide floodplain.*

 AR:  We have changed the wording in the text to make this clearer: although there are still some discrepancies between the channel and the valley centreline from the 30m data, we found that generally the D8 flow routing does not lead to overestimation of valley width from 30 m data as the flow paths are smoother than from the higher resolution data. We have amended the sentence in the text:

> Due to the coarser resolution, extracting a valley centreline was not necessary as the D8 flow routed channel was sufficiently smooth to avoid overestimation of valley floor width at meander bends.

**RC:** *Fig. 12: All regression lines should have confidence bounds associated with them. These could be bootstrapped if autocorrelation in valley floor width is a problem. It will be interesting, however, whether any significant changes between the different creeks can be observed once you account for the uncertainties of the fits.*

 AR:  We have added bootstrapped confidence intervals to each width-area figure (Figs 12, 13, and Fig S1 in the supporting information) as suggested. We bootstrapped the data 1000 times taking a random sample of 50% of the data points in each iteration. We found that these confidence intervals are relatively narrow for each regression, such that it does not affect the results of the analysis as presented in the text.

**RC:** *394: The units of $K_v$ reported here is strange.*

Thanks for catching this - the correct units are m km$^{-0.6}$ as $c_v = 0.3$ for the Appalachian Plateau rivers. We have updated this in the text.